# Advancing Cancer Therapy Predictions with Patient-Derived Organoid Models of Metastatic Breast Cancer

**DOI:** 10.3390/cancers15143602

**Published:** 2023-07-13

**Authors:** Cansu E. Önder, Teresa J. Ziegler, Ronja Becker, Sara Y. Brucker, Andreas D. Hartkopf, Tobias Engler, André Koch

**Affiliations:** 1Research Institute for Women’s Health, University of Tübingen, 72076 Tübingen, Germany; 2Department of Women’s Health, University of Tübingen, 72076 Tübingen, Germany

**Keywords:** metastasis, breast cancer, organoid culture, pleural effusion, ascites, drug response, cancer biology, personalized medicine, patient-derived organoids

## Abstract

**Simple Summary:**

A frequent disabling symptom during metastasized breast cancer is the development of ascites and pleural effusion, which is associated with poor outcomes. Malignant cells in ascites and pleural effusions derived from the primary tumor site indicate the spreading of cancer and can serve as a model for metastatic breast cancer. Therefore, we cultured metastatic cells from six patients with ascites or pleural effusion in a three-dimensional fashion to obtain organoids. The organoids recapitulated the characteristics of metastatic samples, as shown by immunohistochemistry and mutation analysis. Drug assays of organoids were performed to assess individual responses in a personalized manner. Overall, metastatic organoid cultures derived from malignant pleural effusion and malignant ascites demonstrated in vivo-like phenotypes and drug responses. Hence, these metastatic organoids can serve as an accurate model for the investigation of breast cancer progression and therapy predictions.

**Abstract:**

The poor outcome of metastasized breast cancer (BC) stresses the need for reliable personalized oncology and the significance of models recapitulating the heterogeneous nature of BC. Here, we cultured metastatic tumor cells derived from advanced BC patients with malignant ascites (MA) or malignant pleural effusion (MPE) using organoid technology. We identified the characteristics of tumor organoids by applying immunohistochemistry and mutation analysis. Tumor organoids preserved their expression patterns and hotspot mutations when compared to their original metastatic counterpart and are consequently a well-suited in vitro model for metastasized BC. We treated the tumor organoids to implement a reliable application for drug screenings of metastasized cells. Drug assays revealed that responses are not always in accord with expression patterns, pathway activation, and hotspot mutations. The discrepancy between characterization and functional testing underlines the relevance of linking IHC stainings and mutational analysis of metastasized BC with in vitro drug assays. Our metastatic BC organoids recapitulate the characteristics of their original sample derived from MA and MPE and serve as an invaluable tool that can be utilized in a preclinical setting for guiding therapy decisions.

## 1. Introduction

Breast cancer (BC) is the most prevalent type of cancer and one of the leading causes of cancer-associated deaths among women worldwide [1]. The most common kind is invasive BC of no special type (NST), also known as invasive ductal carcinoma (IDC), followed by the special type invasive lobular carcinoma (ILC) [2]. The status of estrogen receptor α (ERα), progesterone receptor (PR), and human epidermal growth factor receptor 2 (HER2) defines the subtype, determines the treatment, and predicts the prognosis [3,4,5]. Breast tumors that express ER and/or PR are categorized as “hormone receptor-positive”, accounting for approximately 65% of cases. The second most common subtype, comprising about 15–20% of cases, involves tumors that overexpress HER2, either alone or in combination with ER/PR. Tumors that do not express ER, PR, or HER2 are referred to as “triple-negative” and constitute around 10–15% of breast cancer cases [6].

Invasive tumor cells can spread through the bloodstream or the lymphatic system to different locations and organs of the body and form metastasis [7]. While primary BC is well understood and has favorable outcomes, metastatic BC remains a challenge. Heterogeneity is one of the causes of metastatic BC and a common reason why therapies are often not successful [8]. Additionally, metastatic BC has a poor prognosis due to persistent and treatment-resistant malignant cells and their impact on vital organs [8].

Due to the rarity of metastatic biopsies, therapy recommendation usually depends on the characteristics of primary tumors. However, it is important to note that primary tumors may not fully reflect the attributes of metastatic lesions, which often exhibit discordance in phenotypic markers such as ER, PR, and HER2 expression [9]. Moreover, there can be alterations in mutational signatures and mutational burden during the progression to metastasis [10]. Therefore, therapy decisions that are based on the characteristics of the primary tumor alone can result in a poor prognosis [11,12].

Malignant ascites (MA) and malignant pleural effusion (MPE) are a widespread concern for cancer patients and describe the presence of malignant cells in the peritoneal and pleural cavity [13,14]. BC is a frequent driver of MA and MPE, with 7% of BC patients developing MPE and 3% developing MA [15,16]. Patients suffering from MPE or MA endure a poor quality of life defined by chest and abdominal pain and difficulty breathing [16,17]. The median survival rate of these patients is relatively low and varies from 5 to 13 months [16,18,19]. MPE and MA, containing metastasized tumor cells, offer opportunities for metastatic biopsies since the fluid can be obtained through a simple puncture to relieve pressure and pain for the patients [20,21]. Consequently, MPE and MA present a beneficial source for the characterization of metastasized BC and the creation of in vitro models to improve therapy outcomes.

The poor prognosis of metastasized BC demonstrates the demand for reliable personalized oncology and the relevance of models representing the characteristics of BC. Cell lines, although widely used in BC research, only represent a small subset of the diversity of BC and are not capable of predicting specific drug responses for patients [19]. Patient-derived xenograft (PDX) models are utilized in translational research and considered crucial for drug development and response studies. Despite their value, high costs and limited efficiency interfere with the use of PDX models [22]. 

Patient-derived BC organoids, cultured in an extracellular matrix, have proven to be an important three-dimensional model for research as they accurately recapitulate the characteristics of their respective origin and can be used for long-term culturing [20]. Organoids of different sources have already been utilized in various methods including drug development and the determination of drug response [23]. Therefore, organoids have been shown to be inexpensive and valuable in vitro models for individualized oncology and high-throughput drug assays [24,25,26].

Since studies addressing BC organoids usually use primary tumor tissue for establishing patient-derived models [24,27,28], we focused exclusively on advanced-stage BC and derived models from MA and MPE, as there is an unmet clinical need in this condition. We further identified the characteristics of these metastatic BC patient-derived organoids (MBC-PDOs) using immunohistochemistry and mutation analysis. Our aim was to utilize these MBC-PDOs to establish a reliable application for drug screenings. Once established, representative organoids can further serve as models to study the most lethal BCs—drug-resistant, metastatic tumors. In the future, research on such patient-derived models could help prolong patients’ survival time and improve their quality of life.

## 2. Materials and Methods

### 2.1. Patient Cohort

All MPE and MA samples were obtained from advanced BC patients treated at the Department of Women’s Health in Tübingen after informed written consent. The study was approved by the Ethics Committee of the Eberhard Karl University of Tübingen (Ethical approval 150/2018BO2 and 288/2022BO2) and is compliant with all relevant ethical regulations regarding research involving human participants. PE (by thoracentesis) and ascites (by paracentesis) from six patients with metastatic BC were collected in sterile containers. For full patient characteristics, see Appendix A.

### 2.2. Culturing 2D Cell Lines

BC cell line MCF-7 was purchased from American Type Culture Collection (Manassas, VA, USA, HTB-22) and served as a control cell line for 3D drug response assays. Cells were cultured in DMEM-FBS (Dulbecco’s Modified Eagle Medium (41965-062), containing 10% FBS (10270-106), 1% Pen/Strep (15140-122); all from Thermo Fisher, Waltham, MA, USA) and incubated at 37 °C and 5% CO_2_. Cells were regularly checked for mycoplasma using a PCR Detection Kit (abm, Richmond, BC, Canada, G238). 

### 2.3. Processing of Pleural Effusions and Ascites

MPE and MA samples were centrifuged at 500× *g* for 10 min. Cell pellets were pooled, and when necessary, red blood cells were lysed with 10 mL of RBC lysis buffer (155 mM NH_4_Cl, 10 mM KHCO_3_, 100 µM Na_2_EDTA in H_2_O, pH 7.4) on ice for 5–10 min. Cells were diluted in DPBS (Dulbecco’s Phosphate Buffered Saline, Pan Biotech GmbH, Aidenbach, Germany, P04-36500) and centrifuged at 500× *g* for 10 min. The final cell pellet was resuspended in AdvDMEM+++ (Advanced DMEM/F-12 (12634-028), 1% Pen/Strep (15140-122), 1x GlutaMAXTM-I (35050-038), 10 mM HEPES (15630-056); all from Thermo Fisher). When applicable, the resulting pellet was subjected to 40 µm and 100 µm cell strainers (Greiner Bio-One GmbH, Frickenhausen, Germany, 42040 and 42000) to obtain various fractions and single-cell solutions. Fractions were used for cryopreservation, culturing organoids, and/or FFPE embedding.

### 2.4. Organoid Culture Setup

For organoid culture setup, the desired amount of cell suspension was mixed with Basement Membrane Extract (BME; Cultrex Reduced Growth Factor Basement Membrane Extract, Type 2 Select, Bio-techne, Minneapolis, MN, USA, 3533-005-02) at a ratio of 30% cell suspension to 70% BME. Twenty µL droplets were plated out on 48-well plates and placed upside down in an incubator (37 °C, 5% CO_2_) to solidify for 30 min. BC culture medium (BCM; Appendix A, composition previously described [24,27]) was added to each well and renewed every 3–4 days. Cells were incubated at 37 °C and 5% CO_2_, and pictures were taken regularly with an EVOS M7000 microscope (Thermo Fisher).

### 2.5. Passaging of Organoid Cultures 

MBC-PDOs were passaged every 7 to 21 days, depending on organoid size and density. Organoids were recovered from the wells by resuspending the BME droplets in ice-cold DPBS containing 5 µM Y-27632 (DPBS/Y-27632). The organoid suspension was centrifuged at 500× *g* for 10 min, and the supernatant was removed. The BME-organoid pellet was enzymatically dispersed with 1 mL of TrypLE Express Enzyme (1X; Thermo Fisher, 12604013) at 37 °C in a water bath for 5 min. The suspension was then centrifuged at 500× *g* for 10 min, and the supernatant was removed. For further culture, the desired amount of cell pellet was resuspended in AdvDMEM+++ and mixed with BME at a ratio of 30% cell suspension to 70% BME and cultured as described above. To stock organoids, passaged cells were cryopreserved in Recovery Cell Culture Freezing Medium (Thermo Fisher, Waltham, MA, USA, 12648010) and stored in cryovials in liquid nitrogen.

### 2.6. Three-Dimensional Drug Screening

For the heat-map assays, specific concentrations of the drugs were used. For the dose-dependent assays, dilution series of various drugs were prepared in BC medium. Drugs applied in the assays are listed in Appendix A. 

MBC-PDOs were passaged as follows: Two BME droplets containing organoids were resuspended in 500 µL of TrypLE, incubated at 37 °C for 10 min, and afterward thoroughly pipetted up and down to disrupt organoids into single cells. Cells were shortly spun down and resuspended in AdvDMEM+++. Cells were counted using the Bio-Rad TC20 Automated Cell Counter according to the manufacturer’s protocol. To seed 1000 cells/well, the required cell number of each line was resuspended in BME in a ratio of 30% cell suspension and 70% BME. Three µL of this suspension, containing 1000 cells, was seeded in each well of white flat-bottom 96-well plates (Thermo Fisher, 136102). Plates were turned upside down and incubated at 37 °C and 5% CO_2_ for 30 min. Finally, drug dilutions in BCM were added to the wells. 

After four days of treatment, relative cell viability was determined using CellTiter-Glo 3D Cell Viability Assay reagent (Promega, Madison, WI, USA, G9682). The reagent was mixed 1:1 with the same volume of AdvDMEM+++. The content of the wells was removed, and 100 µL of reagent mixture (1:1 mix with AdvDMEM+++) was added to the wells. Then, plates were incubated at room temperature on an orbital plate shaker for 15 min (900 rpm). After a 10 min resting phase, luminescence values were read using a Varioskan LUX (Thermo Fisher). Assays were performed in technical triplicates. For the drug response curves, results were normalized to the lowest drug concentration. For the heat-map assay, results were normalized to viability in 0.1% DMSO (set as 100% cell viability).

### 2.7. Paraffin Sections and Immunohistochemistry of Tissue and MBC-PDOs

Samples were fixed in 4% formaldehyde, followed by dehydration, paraffin embedding, sectioning (2.5 µm), and standard hematoxylin and eosin (H&E) staining. Immunohistochemistry (IHC) was performed using the ZytoChem Plus HRP Polymer Kit (Zytomed Systems GmbH, Bargteheide, Germany, POLHRP-100) and DAB Substrate Kit (Zytomed Systems GmbH, DAB057) according to the manufacturer’s protocol. IHC was performed using antibodies listed in Appendix A. Images were captured with ScanScope (Leica, Wetzlar, Germany) or EVOS M7000 microscope (Thermo Fisher) and processed with Aperio ImageScope (version 12.4.6.5003), Image J (version 1.53e), and Photoshop (version 13.0.1).

### 2.8. Mutation Analysis

Four different hotspot regions were investigated for mutation analysis. Primers were designed using NCBI (see Appendix A). The reverse primer for *PIK3CA* E542 and E545 mutation was designed based on a previous publication, to eliminate the amplification of a pseudogene [29]. Nucleic acids were extracted using the Quick-DNA/RNA miniprep kit (Zymo Research, Irvine, CA, USA, D7001) according to the manufacturer’s protocol. DNA concentration was determined using a Varioskan LUX (Thermo Fisher).

Polymerase chain reaction (PCR) was performed by preparing 20 µL reactions with 1x Phusion HF buffer, 200 µM dNTPs, 0.5 µM of forward and reverse primer (Appendix A), 40 ng of template DNA, 0.4 units Phusion DNA polymerase, and nuclease-free water up to 20 µL. After an initial denaturation at 98 °C for 60 s, PCR was run for 34 cycles: denaturation at 98 °C for 10 s, annealing at 62 °C for 30 s, elongation at 72 °C for 60 s. Final elongation was performed at 72 °C for 10 min. PCR products were analyzed via agarose gel electrophoresis and purified using the QIAquick PCR Purification Kit (QIAGEN GmbH, Hilden, Germany, 28106) according to the manufacturer’s instructions. Amplified DNAs were sequenced by Eurofins Genomics Germany GmbH, applying Sanger sequencing. Results were analyzed and aligned using SnapGene Viewer (version 6.0.2).

## 3. Results

### 3.1. Establishing a Biobank of Metastasized BC Organoids Derived from Pleural Effusion and Ascites

To generate suitable models for metastasized BC, pleural effusion and ascites were obtained from patients with metastasized BC. Isolated tumor cells were seeded in BME droplets and cultured in BC medium as previously described [24]. Figure 1A shows an example of an established organoid line (MBC-PDO #07) that has been cultured for at least 10 passages. The morphology of the organoids was maintained in high passage (P) numbers. We successfully established organoid lines with various morphologies, including grape-like structures, dense and massive organoids, and lines with smooth or rough structures (Figure 1B).

BC is known to be classified into specific subtypes based on histological characteristics and receptor status (ERα, PR, and HER2) [30]. The most common BC subtypes are invasive ductal carcinoma (IDC) of no special type (NST) and invasive lobular carcinoma (ILC) [31]. Both types are represented in our organoid models (Appendix A). The dominating receptor status is hormone receptor-positive/HER2-negative BC, which is also represented by the MBC-PDO lines (Table 1).

### 3.2. Immunohistochemical Characterization of Patient-Derived Organoids

The receptor status of BC influences the type of therapy the patients receive and is an indicator for prognosis. For example, triple-negative BC is associated with a poor prognosis as it lacks hormone receptors and has limited therapy options, while hormone receptor (ERα and PR)-positive tumors can be treated with endocrine therapy. HER2-positive BC can be targeted with antibodies and inhibitors.

We studied the suitability of organoid lines as a model for metastasized BC and their conservation of histological characteristics through H&E and IHC staining of pleural samples and the corresponding organoid lines. The phenotype and expression pattern of the liver metastasis, MPE, and the corresponding MPE organoids of MBC-PDO #03 were compared. The corresponding patient was diagnosed with primary NST in 2008 and developed PE 13 years later. To confirm that expression patterns are retained in organoids, organoid samples of MBC-PDO #03 were taken at P3 for FFPE embedding and IHC staining (Figure 2). H&E staining revealed comparable histology of liver metastasis, pleural cells, and organoids. IHC staining of nuclear hormone receptors ER and PR demonstrated positive signals in all three samples (Figure 2). Membranous receptor HER2 could be observed in some of the cells. According to pathology, HER2 expression had an IHC score of 2+ (out of 3+), which was represented in the patient-derived organoids. Thus, organoids of MBC-PDO #03 not only preserved the expression patterns of pleural cells, but also had the same receptor status as the respective liver metastasis.

IHC staining of additional markers (epithelial marker EpCAM, GATA3, tumor suppressor protein p53, and cadherin-1) in the pleural effusion sample revealed positive signals, which were equally preserved in the derived organoid culture (Appendix A). Ki-67 staining was applied to compare the proliferation of organoids. Only a small number of Ki67-positive cells were observed for MBC-PDO #02, #03, and #04, whereas MBC-PDO #06 and #07 presented a higher number of Ki67-positive cells (Appendix A). This is in accordance with the proliferation rate we observed in the organoid lines. An overview of the IHC staining results of all lines is provided in Table 1. In most cases, organoids preserved the expression patterns of the original cells. One exception was observed in MBC-PDO #04: no ERα expression was detected in the organoids, while pleural cells displayed low levels of ERα expression. In conclusion, the receptor status of tumor organoids correlates with the expression patterns of their metastatic counterpart, suggesting that these organoids could serve as a well-suited in vitro model for metastasized BC. 

### 3.3. Hotspot Mutation Analysis of Patient-Derived Organoid Lines

Tumorigenesis in BC cells can be driven by both the phosphoinositide 3-kinase (PI3K) and mitogen-activated protein kinase (MAPK) signaling pathway [32]. Activation of these pathways can sometimes be traced back to mutations of key proteins. Mutations in *AKT1* and *PIK3CA* can lead to abnormal protein production and have been associated with the activation of downstream processes (proliferation, survival, etc.) and increased risk of BC [33,34,35].

A missense mutation of *AKT1* leading to the exchange of glutamic acid (E) to lysine (K) in amino acid position 17 has been observed in approx. 6% of all BC cases [36]. Mutations in *PIK3CA* are one of the most common mutations in BC and have been observed in more than 30% of BC cases [35,37]. The three most common mutation hotspots in this gene are E542 (11%), E545 (17%), and H1047 (35%) [38].

Sanger sequencing of the hepatic metastasis of MBC-PDO #03 revealed a *PIK3CA* H1047R mutation that was also detected in the MPE as well as the corresponding organoid model (Table 1 and Appendix A). Whereas none of the samples presented with a *PIK3CA* E542 mutation, MBC-PDO #05 and MBC-PDO #06 had an E545 hotspot mutation that was also present in the primary tumor of the corresponding patients. A mutation in *AKT1* (E17K) was found in the MPE of MBC-PDO #04 and #07 as well as the primary tumor samples. The MPEs of MBC-PDO #02 and MBC-PDO #05 could not be tested due to insufficient sample material from the initial MPE. MBC-PDO #02 was also the only organoid line with no hotspot mutation detected. Thus, five out of six organoid lines carried a mutation in either *AKT1* or *PIK3CA*.

Mutation analysis of hotspot gene mutations showed that the organoids retained mutations from initial MPE (and often the primary tumor) and that derived organoids represent the metastasized tumors. These findings indicate that the organoid lines could be useful for downstream applications like drug response assays.

### 3.4. Drug Response Assays

To examine the potential of organoids derived from MPE and MA as in vitro models of BC metastasis, we assessed their suitability for use in drug response assays. We selected drugs that target the HER signaling pathway, cyclin-dependent kinases 4 and 6 (CDK4/6), poly ADP ribose polymerases (PARPs), DNA synthesis, and β-tubulin (Figure 3A). Organoid lines were treated with various drugs at specific concentrations, and the results are presented in a heat map (Figure 3B). Serial dilution drug assays of Lapatinib, Alpelisib, Capivasertib, Abemaciclib, and Paclitaxel were performed to generate dose–response curves and calculate the half-maximal inhibitory concentrations (IC50) (Figure 3C).

MBC-PDO #03, which has medium levels of HER2 expression (Figure 2), showed a strong response to the HER2 pathway inhibitor Lapatinib. Despite having the H1047R hotspot mutation in the *PIK3CA* gene (Table 1), which could potentially activate the corresponding signaling pathway, MBC-PDO #03 did not respond positively to drugs targeting the PI3K pathway (Figure 3). As tumorigenesis in BC cells can be driven by the PI3K and MAPK signaling pathways, we examined if the PI3K signaling cascade was activated. IHC staining for the activating phosphorylation site Ser473 of AKT showed that p-AKT was absent in MBC-PDO #03 compared to the positive staining in MBC-PDO #06, suggesting that the PI3K pathway was not activated in MBC-PDO #03 (Appendix A). This demonstrates the value of using functional in vitro drug assays in combination with p-AKT IHC stainings in patient-derived organoids.

MBC-PDO #05 carries the *PIK3CA* E545K hotspot mutation and reacted most sensitively to AKT inhibitor Capivasertib, followed by the other PI3K and AKT drugs. We did not have sufficient organoids from MBC-PDO #05 to fix and perform IHC staining. However, the patient would likely have profited from treatment with PI3K or AKT inhibitors like Alpelisib, instead of the actual therapy with Paclitaxel combined with HER2-targeting Trastuzumab and Pertuzumab, followed by CDK4/6 inhibitor Abemaciclib combined with Trastuzumab.

MBC-PDO #07, which is positive for p-AKT (Table 1) and carries an *AKT1* E17K mutation, responded most sensitively to the AKT inhibitor Capivasertib. This line was also sensitive to Gemcitabine. Interestingly, the patient of MBC-PDO #07 had previously been treated with Paclitaxel and Abemaciclib, and both drugs did not affect the cell viability of MBC-PDO #07, suggesting a resistance to these drugs. Furthermore, the corresponding patient was diagnosed with a *BRCA1/2* deletion in the breast tumor sample and therefore had previously received treatment with the PARP inhibitor Olaparib. Nonetheless, the organoid line was resistant to Olaparib. Consequently, the patient may have benefited from treatment with AKT inhibitors or Gemcitabine.

In summary, our results emphasize the relevance of combining IHC staining and mutation analysis with in vitro drug assays in order to move a step closer to personalized BC therapy. For instance, even in cases where the HER2 signal is negative or low, it may be beneficial for some patients to receive HER2-targeting therapy like Lapatinib.

## 4. Discussion

The poor outcome of metastasized BC underscores the need for reliable personalized oncology and the importance of models recapitulating the characteristics of MBC [8]. Two of the most used approaches involve cell lines as well as PDX models. While cell lines are readily available but often do not recapitulate drug responses in patients [22], PDX models are good in therapy predictions but are expensive and need a long time for their establishment [22]. Recent years have seen the rise of organoid technology for personalized oncology with an enormous potential for clinical applications [24,25]. However, while organoids are often established from primary tumor specimens obtained during resection of the tumor, metastases are often underrepresented due to insufficient bioavailability of specimen material. Biopsies are often small and are still widely used for pathological characterization only. Drug efficacy screenings are therefore mainly performed on organoids derived from treatment-naïve primary tumors rather than on already heavily pre-treated metastatic samples. These pose the disadvantage of not being representative of metastatic disease and often do not help the patients in need. Here, we were trying to close this gap by utilizing MPE and MA from breast cancer patients as a source for the generation of such metastatic models. One great advantage of this approach is the facile accessibility to metastasized tumor cells, which can easily be obtained by clinical routine procedures, such as thoracentesis and paracentesis. BC patients with MA or MPE can suffer from a shortness of breath, discomfort, and chest or abdominal pain, which collectively lead to a reduced quality of life. Hence, one of the main objectives is to improve patients’ quality of life and prolong their survival time by reducing MPE and MA.

This study shows the feasibility of culturing metastatic tumor cells derived from advanced BC patients with MA or MPE utilizing organoid technology. The derived tumor organoid lines could be cultured for extended passages (Figure 1) and presented characteristics of their respective source materials (see IHC and mutation analysis; Figure 2, Table 1). The collection of tumor organoid lines represents the different classifications of BC regarding subtype, histological features, and receptor status, which to date are key factors in therapy recommendation. However, the main objective here was to utilize these models for drug screenings and therapy predictions.

In our study, drug assays revealed that responses are not always in accord with expression patterns, pathway activation, and hotspot mutations. This emphasizes the value of personalized drug assays, in combination with the characterization of expression patterns and mutation status. For instance, HER2-targeting drugs affected some of the MBC-PDO lines with zero to low levels of HER2 (Figure 3, Table 1). It is noteworthy that some of these drugs target multiple receptors, including EGFR, highlighting potential benefits in clinical applications, despite a negative HER2 status. In fact, clinical trials with Afatinib in HER2-negative BC have shown some effect on tumor progression [39,40].

PI3K- and AKT-targeting drugs affected most hotspot mutated tumor organoid lines, as anticipated. Overall, AKT inhibitor Capivasertib showed the highest effect among all drugs—four of six MBC-PDO lines responded sensitively. According to a study, alterations in the PI3K-AKT signaling pathway have been linked to higher levels of p-AKT, and responsiveness to AKT inhibitor Ipatasertib [41].

Here, however, one *PIK3CA*-mutated line showed less sensitivity, as the IHC staining of p-AKT indicated an inactive PI3K cascade in those organoids (Appendix A, MBC-PDO #03). This stresses the necessity of including a biomarker such as p-AKT in clinical diagnostics to guide therapy decisions. According to the mentioned study, triple-negative BC tumors with phosphorylated AKT levels were associated with enriched clinical benefit of AKT inhibitor Ipatasertib regardless of the absence of PI3K-AKT alterations [41].

According to the SOLAR-1 study, the combination of Alpelisib and Fulvestrant resulted in a higher median overall survival for patients with *PIK3CA*-mutated, HR+, HER2- advanced BC [42]. Here, we have demonstrated that IHC staining of HER2 and mutation analysis of *PIK3CA* or *AKT1* genes are relevant but not sufficient for a proper therapy recommendation. Drugs targeting HER2, such as Lapatinib, can also be beneficial for patients with HER2-, advanced BC. In contrast, patients with an altered *PIK3CA* or *AKT1* can, but do not necessarily, profit from a treatment targeting the PI3K-AKT pathway, as we have shown for MBC-PDO #03 (Figure 3 and Table 1). We propose that in addition to receptor determination and mutational diagnosis, staining of p-AKT can help with the understanding of whether the corresponding pathway is activated and if targeting PI3K or AKT could be reasonable.

CDK4/6 inhibitors and mTORC1 inhibitor Everolimus are widely used in the clinic for hormone receptor-positive and HER2-negative BC patients [43,44,45]. In particular, patients with endocrine-resistant BC may profit from these drugs. In our drug assays, however, the response to CDK4/6 inhibitors and Everolimus did not correlate with the receptor status of the tumor organoids. One possible explanation would be that some MBC-PDOs acquired resistance towards these drugs upon clinical treatment. Similar effects have already been observed in previous studies [46]. These results repeatedly suggest that personalized drug response assays can be beneficial for individual cases, and new biomarkers could help identify patients that would profit from CDK4/6 inhibitors and Everolimus.

In summary, the discrepancy between characterization and functional testing underlines the relevance of linking IHC stainings and mutational analysis of metastasized BC with in vitro drug assays, to improve personalized BC therapy. To make organoid-based drug screening practical in a clinical setting, it is crucial to have a fast turnaround time for organoid culture establishment and drug response assessment, preferably within the same time interval as the treatment. Here, we were able to culture and screen tumor organoid lines within two weeks. To further confirm the accuracy of metastatic organoids in predicting drug response, it would be beneficial to conduct a study in which metastatic organoids derived from patients with MA or MPE are screened with drugs used in the clinic. Consequently, the results can then be aligned with the patients’ response to the treatment.

We established six MBC-PDO lines and discovered that with more models, hypotheses and biomarkers can be thoroughly investigated in our setup. Generating a larger biobank of metastatic BC organoids will help with the understanding of correlations between a patient’s genetic as well as proteomic makeup and drug responses to identify sensitivity and resistance across BC subtypes. With a larger collection of drug screening data, it could be possible to create therapy plans based on the characteristics of the organoid subtypes.

## 5. Conclusions

In conclusion, our metastatic BC organoids recapitulate the characteristics of their original sample derived from MA and MPE and serve as an invaluable model that can be utilized in a preclinical setting for guiding therapy decisions. Further research is necessary to prove the validity of these models for clinical relevance. Since the minority of BC patients develop MA or MPE, alternative sources for liquid biopsies, e.g., blood samples, should be standardized in pathological and experimental applications to maximize the patient numbers that are included in these studies. Additionally, correlating drug screenings with genomic and proteomic analysis can be beneficial in finding new biomarkers for drug response in the future.

## Figures and Tables

**Figure 1 cancers-15-03602-f001:**
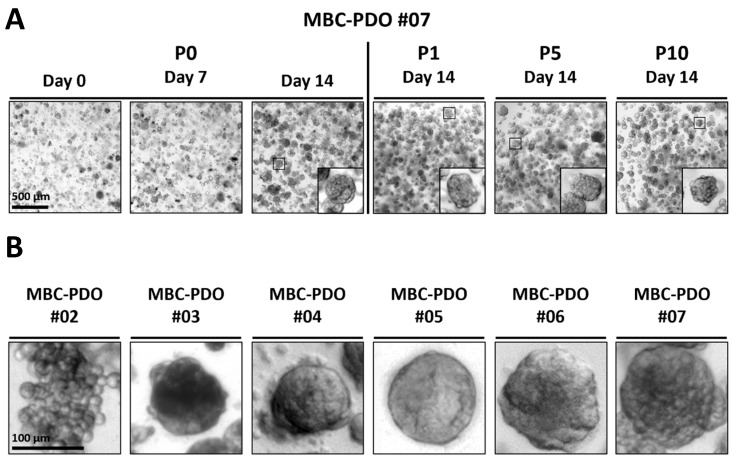
Establishing a biobank of breast cancer (BC) organoids derived from pleural effusion and ascites. (**A**) Brightfield images of metastatic BC patient-derived organoid (MBC-PDO) #07 in passage (P)0, P1, P5, and P10. Morphology was preserved throughout the whole culturing period. Scale bar: 500 µm. (**B**) Brightfield images of organoid lines displaying different phenotypes. The first image with grape-like morphology illustrates the ILC sample MBC-PDO #02, and images with dense, smooth, and rough structures illustrate NST samples of MBC-PDO #03–#07. Scale bar: 100 µm.

**Figure 2 cancers-15-03602-f002:**
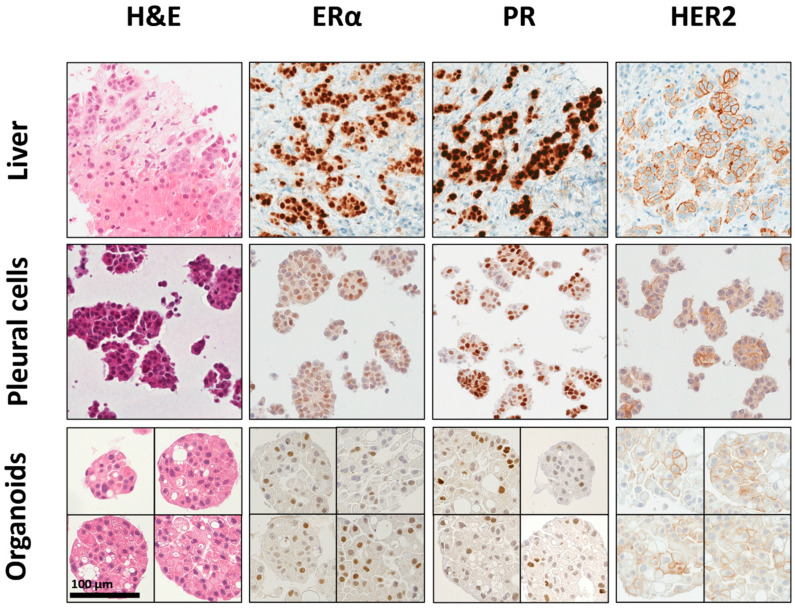
Histological characterization of pleural BC cells and organoids derived from pleural effusion. Histology (H&E staining) and receptor status of metastasized liver tissue, pleural cells, and organoids (P3) of MBC-PDO #03 are shown. Organoids consist exclusively of epithelial cells, while tissues often show tumor epithelium ringed by mesenchymal cells. The receptor status is maintained in the organoids. Scale bar: 100 µm.

**Figure 3 cancers-15-03602-f003:**
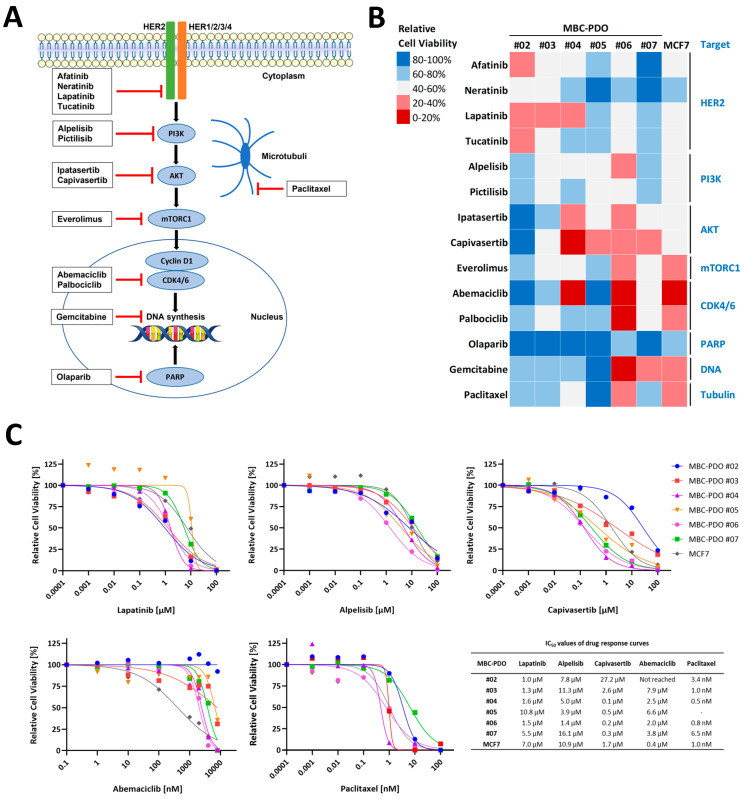
In vitro drug response assays of organoid lines. (**A**) Overview of drugs used in the assays and their targets. Inhibitors Afatinib, Lapatinib, Neratinib, and Tucatinib target the HER2 receptor intracellularly. Alpelisib and Pictilisib target PI3K, while Ipatasertib and Capivasertib aim for AKT. Everolimus inhibits mTORC1, whereas Abemaciclib and Palbociclib target CDK4/6 in the nucleus. Olaparib, which is administered to BC patients with a *BRCA1/2* mutation/deletion, inhibits PARP. Gemcitabine enters the cell through nucleoside transporters and is phosphorylated in three steps into Gemcitabine triphosphate, which is incorporated into the DNA and leads to the termination of DNA synthesis. Paclitaxel binds and stabilizes β-tubulin, upon which depolymerization is blocked. This affects the mitotic spindle assembly, chromosome segregation, and mitosis. (**B**) Heat map of relative cell viability of organoids treated with various drugs at specific concentrations for 4 days (10 µM Afatinib, 1 µM Neratinib, 4 µM Lapatinib, 10 µM Tucatinib, 10 µM Alpelisib, 1 µM Pictilisib, 1 µM Ipatasertib, 1 µM Capivasertib, 100 nM Everolimus, 4 µM Abemaciclib, 10 µM Palbociclib, 10 µM Olaparib, 100 nM Gemcitabine, 10 nM Paclitaxel). Values were normalized to control treatment with 0.1% DMSO (set to 100%). Red fields indicate sensitive responses, while blue fields present less-responding lines. Experiments were performed in technical triplicates. (**C**) Drug response curves depict organoid viabilities after 4 days of treatment with Lapatinib, Alpelisib, Capivasertib, Abemaciclib, and Paclitaxel. Error bars representing standard deviation (SD) of three independent experiments were removed for better presentation. IC50 values of drug response curves are shown in the adjacent table.

**Table 1 cancers-15-03602-t001:** Results of immunohistochemistry (IHC) staining signals of pleural cells and/or organoid lines, and mutation analysis of hotspots in *AKT1* and *PIK3CA*. Abbreviations neg.: negative signal, pos.: positive signal, WT: wild type, E: glutamic acid, K: lysine, H: histidine, R: arginine.

MBC-PDO		ERα	PR	HER2	*AKT1*	*PIK3CA*	p-AKT
#02	Organoids	neg.	neg.	neg. *	WT	WT	neg.
#03	PE	pos.	pos.	neg. *	WT	H1047R †	-
	Organoids	pos.	pos.	neg. *	WT	H1047R †	neg.
#04	PE	pos. *	neg.	neg.	E17K ◊	WT	-
	Organoids	neg.	neg.	neg.	E17K ◊	WT	pos.
#05	PE	pos.	pos.	neg.	-	-	-
	Organoids	-	-	-	WT	E545K †	-
#06	PE	pos. *	neg.	neg.	WT	E545K †	-
	Organoids	pos.*	neg.	neg.	WT	E545K †	pos.
#07	PE	pos. *	neg.	neg.	E17K ◊	WT	-
	Organoids	pos. *	neg.	neg.	E17K ◊	WT	pos.

* Samples showed a weak positivity, ◊ homozygous mutation, † heterozygous mutation.

## Data Availability

The data demonstrated in this study are available on request from the corresponding author.

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
