# Peer review of "Advancing Cancer Therapy Predictions with Patient-Derived Organoid Models of Metastatic Breast Cancer"

_cancers, 2023, doi:10.3390/cancers15143602_

Round 1
Reviewer 1 Report
The present study was undertaken to investigate a new idea for drug screening in metastatic breast cancer. The authors have cultured metastatic tumor cells derived from advanced breast cancer patients with malignant ascites or malignant pleura effusion using organoid technology. Then, they have identified the characteristics of tumor organoids and used them as models to study the most lethal breast cancers. They have finally proposed that the research on such patient-derived models could help prolong patients’ survival time and improve their quality of life.
The manuscript is well-organized, the methods are well-described, and the results are well-presented and discussed.
Author Response
Dear reviewer,
We highly appreciate your positive feedback. We thank you for your precious time in reviewing our paper. For your information, we have focused on addressing minor details that were brought up by the other reviewers.
Thank you very much!
Reviewer 2 Report
In this paper, authors cultured metastatic tumor cells derived from advanced BC patients with malignant ascites (MA) or malignant pleura effusion (MPE) using organoid technology. They identified the characteristics of tumor organoids applying immunohistochemistry and mutation analysis. Tumor organoids preserved their expression patterns and hotspot mutations when compared to their original metastatic counterpart and are consequently a well-suited in vitro model for metastasized BC. They treated the tumor organoids to implement a reliable application for drug screenings of metastasized cells. Authors did a good work and interested for the readers. Following review comments are recommended, and the authors are invited to explain and modify.
1 The abstract section is inconsistent and does not reflect the main contributions of the manuscript. The authors should rewrite the abstract section to mention the primary contributions, experimental results, and global implications.
2 Novelty is confusing. A highlight is required. The main contributions of the manuscript are not clear. The main contributions of the article must be very clear and would be better if summarize them into 3-4 points at the end of the introduction.
3 Introduction section needs to be improve. An introduction is an important road map for the rest of the paper that should be consist of an opening hook to catch the researcher's attention, relevant background study, and a concrete statement that presents main argument but your introduction lacks these fundamentals, especially relevant background studies. This related work is just listed out without comparing the relationship between this paper's model and them; only the method flow is introduced at the end; and the principle of the method is not explained. To make soundness of your study must include these latest related works. Authors also need to justify the importance of their article and cite all of them to make a critical discussion that makes a difference from others' work.
I (2022). Calcium Homeostasis in Parkinson’s Disease: From Pathology to Treatment. Neurosci. Bull. 38, 1267–1270. https://doi.org/10.1007/s12264-022-00899-6
II (2018). RSL3 Drives Ferroptosis Through GPX4 Inactivation and ROS Production in Colorectal Cancer. Frontiers in Pharmacology, 9. doi: 10.3389/fphar.2018.01371
III (2021). Small molecule DNA-PK inhibitors as potential cancer therapy: a patent review (2010–present). Expert Opinion on Therapeutic Patents, 31(5), 435-452. doi: 10.1080/13543776.2021.1866540
IV (2021). Atractylenolide I enhances responsiveness to immune checkpoint blockade therapy by activating tumor antigen presentation. The Journal of Clinical Investigation, 131(10). doi: 10.1172/JCI146832
4 “For organoid culture setup, the desired amount of cell suspension was mixed with Basement Membrane Extract at a ratio of 30% cell suspension to 70% BME”, what is logic to choose this ratio?
5 “For the heat-map assays, specific concentrations of the drugs were used”, what is that specific concentrations of the drugs?
6 “Four different hotspot regions were investigated for mutation analysis”, what is the importance of four different hotspot regions?
7 Authors should mention the implementation challenges.
Minor editing of English language required.
Author Response
Dear reviewer,
We highly appreciate your feedback on our manuscript. We thank you for your precious time in reviewing our paper. Please find below a point-by-point response to the suggestions you have raised.
1 The abstract section is inconsistent and does not reflect the main contributions of the manuscript. The authors should rewrite the abstract section to mention the primary contributions, experimental results, and global implications.
Dear reviewer, we do not fully agree with this suggestion. In line 22-24 we introduce the methodology (organoid technology and patient material from metastatic breast cancer patients). We follow up on this with experimental results in line 24-31 and give an outline for clinical implications in line 31-33. In our opinion, the abstract already reflects all the points you want to have included in an abstract, and so we did not introduce any changes.
2 Novelty is confusing. A highlight is required. The main contributions of the manuscript are not clear. The main contributions of the article must be very clear and would be better if summarize them into 3-4 points at the end of the introduction.
Dear reviewer, thanks for the suggestion. We have addressed these points in the last section of our introduction (line 88-98).
3 Introduction section needs to be improve. An introduction is an important road map for the rest of the paper that should be consist of an opening hook to catch the researcher's attention, relevant background study, and a concrete statement that presents main argument but your introduction lacks these fundamentals, especially relevant background studies. This related work is just listed out without comparing the relationship between this paper's model and them; only the method flow is introduced at the end; and the principle of the method is not explained. To make soundness of your study must include these latest related works. Authors also need to justify the importance of their article and cite all of them to make a critical discussion that makes a difference from others' work.
I (2022). Calcium Homeostasis in Parkinson’s Disease: From Pathology to Treatment. Neurosci. Bull. 38, 1267–1270. https://doi.org/10.1007/s12264-022-00899-6
II (2018). RSL3 Drives Ferroptosis Through GPX4 Inactivation and ROS Production in Colorectal Cancer. Frontiers in Pharmacology, 9. doi: 10.3389/fphar.2018.01371
III (2021). Small molecule DNA-PK inhibitors as potential cancer therapy: a patent review (2010–present). Expert Opinion on Therapeutic Patents, 31(5), 435-452. doi: 10.1080/13543776.2021.1866540
IV (2021). Atractylenolide I enhances responsiveness to immune checkpoint blockade therapy by activating tumor antigen presentation. The Journal of Clinical Investigation, 131(10). doi: 10.1172/JCI146832
Dear reviewer, you raised several concerns in this section. Regarding ‘relevant background studies’: No one in the field is using metastatic breast cancer samples for the derivation of patient-derived organoid models. We have cited all relevant studies using primary breast tumor tissue (References 24, 27, 28) and comment e.g. in line 55-62, and 88-90, that this poses a drawback when translating these findings to advanced breast cancer. We hope this is sufficient for you.
‘only the method flow is introduced at the end; and the principle of the method is not explained.’
We are not explaining the methodology of organoid derivation since this would go beyond the scope of an introduction found in a ‘Cancers’ Research Article. We are citing highly acclaimed reviews (References 25 and 26) that the interested reader can use for a more in-depth study of organoid technology. Furthermore, in the method and result section you can find more details of how we are generating these organoids.
Furthermore, you are suggesting to include four citations into the introduction and we have obvious concerns in doing so.
Citation 1: (2022). Calcium Homeostasis in Parkinson’s Disease: From Pathology to Treatment.
This article has no relation to breast cancer nor patient-derived organoids. It is about Parkinson’s Disease. We have not included this citation.
Citation 2: (2018). RSL3 Drives Ferroptosis Through GPX4 Inactivation and ROS Production in Colorectal Cancer.
This article has no relation to breast cancer nor patient-derived organoids. Furthermore, we are not working with Ferroptosis or mention ROS production. We have not included this citation.
Citation 3: (2021). Small molecule DNA-PK inhibitors as potential cancer therapy: a patent review (2010–present).
This article has no relation to breast cancer nor patient-derived organoids. Furthermore, DNA-PK inhibitors have no implication in breast cancer treatment. We have not included this citation.
Citation 4: (2021). Atractylenolide I enhances responsiveness to immune checkpoint blockade therapy by activating tumor antigen presentation.
This article has no relation to breast cancer nor patient-derived organoids. Furthermore, we are not mentioning immune checkpoint blockade therapy. We have therefore not included this citation.
4 “For organoid culture setup, the desired amount of cell suspension was mixed with Basement Membrane Extract at a ratio of 30% cell suspension to 70% BME”, what is logic to choose this ratio?
Dear reviewer, we are using a ratio of 30% cell suspension to 70% BME as this the standard ratio for organoid culture setup found in numerous papers in the field of (breast cancer) organoids (see for example PubmedID: 29224780 and 33692550).
5 “For the heat-map assays, specific concentrations of the drugs were used”, what is that specific concentrations of the drugs?
Dear reviewer, thank you for pointing out the missing values. We have included the missing values now in the figure legend of Figure 3 (see line 347 – 349).
6 “Four different hotspot regions were investigated for mutation analysis”, what is the importance of four different hotspot regions?
Dear reviewer,
Tumorigenesis in BC cells can be driven by both the Phosphoinositide 3-kinase (PI3K) and mitogen-activated protein kinase (MAPK) signaling pathway. Activation of these pathways by mutations in AKT1 and PIK3CA have been associated with the activation of downstream processes (proliferation, survival, etc.) and increased risk of breast cancer. Furthermore, these mutations are found in a high prevalence and have been therefore investigated here.
We are stating the importance of the hotspot mutations in PIK3CA and AKT1 in the beginning of section 3.3 (see lines 269 – 280) with numerous citations.
7 Authors should mention the implementation challenges.
Dear reviewer, we have included a statement of the implementation challenges into clinical practice in the conclusion section (line 447 - 451).
We thank you again for your precious time and hope the implemented changes are satisfactory for you!
Reviewer 3 Report
The authors proposed to culture tumor cells derived from MA and PE of advanced BC patients applying organoid technology. Furthermore, they identified the characteristics of these metastatic BC patient-derived organoids (MBC-PDOs) using immunohistochemistry and mutation analysis. Their aim was to utilize these MBC-PDOs to establish a reliable application for drug screenings.
The study is well written, is easy to follow and covers an hot topic. In fact, once established, patient-derived models organoids can could help prolong patients’ survival time and improve their quality of life. So, I have no futher comments against the manuscript.
Author Response

(The authors gave the same response as above.)

Reviewer 4 Report
I congratulate the authors of the idea and the research done.
Author Response

(The authors gave the same response as above.)

Reviewer 5 Report
Comments to the Author
Önder and co-workers present an interesting study on breast cancer. They cultured metastatic cells from six patients with ascites or pleural effusion in a 3-dimensional fashion to obtained organoids. Tumor organoids preserved their expression patterns and hotspot mutations when compared to their original metastatic counterpart and are consequently a well-suited in vitro model for metastasized BC.
The autors conclude that these metastatic organoids can serve as an accurate model for the investigation of breast cancer progression and therapy predictions.
Generally the technical part of this work seems to be well conducted and performed. The procedures and techniques used are standard and appear appropriate.
I believe that the results are of clinical relevance.
Author Response

(The authors gave the same response as above.)

Reviewer 6 Report
This is a good paper that can be published after minor revision.
1. BC is known to be classified into specific subtypes based on histological characteris tics and receptor status. Could the authors make it more clear that what are the dominant histological characteris tics and receptor status?
2. The poor prognosis of metastasized BC demonstrates the demand for reliable personalized oncology and the relevance of models representing the characteristics of BC. Please explain more in detail why it is poor prognosis for metastasized BC.
Author Response
Dear reviewer,
We highly appreciate your positive feedback. We thank you for your precious time in reviewing our paper. Regarding your suggestions we made the following changes: We explain in more detail the dominant histological characteristics and receptor status of BC (line 44-48 and 214-216). The second improvement explains why metastasized BC has a poorer prognosis (line 51-62).
We hope these changes are sufficient for you. Thank you very much!
Round 2
Reviewer 2 Report
Accepted.
Minor editing of English language required.